# Bio-Skin: A Cost-Effective Thermostatic Tactile Sensor with Multi-Modal Force and Temperature Detection

Haoran Guo, Haoyang Wang, Zhengxiong Li, *Member, IEEE,* Lingfeng Tao, *Member, IEEE*

*Abstract*— Tactile sensors can significantly enhance the perception of humanoid robotics systems by providing contact information that facilitates human-like interactions. However, existing commercial tactile sensors focus on improving the resolution and sensitivity of single-modal detection with high-cost components and densely integrated design, incurring complex manufacturing processes and unaffordable prices. In this work, we present Bio-Skin, a cost-effective multi-modal tactile sensor that utilizes single-axis Hall-effect sensors for planar normal force measurement and bar-shape piezo resistors for 2D shear force measurement. A thermistor coupling with a heating wire is integrated into a silicone body to achieve temperature sensation and thermostatic function analogous to human skin. We also present a cross-reference framework to validate the two modalities of the force sensing signal, improving the sensing fidelity in a complex electromagnetic environment. Bio-Skin also demonstrates its performance metrics—including noise-to-range ratio, sampling rate, and measurement range—comparable to current commercial products, with one-tenth of the cost. The sensor's real-world performance is evaluated using an Allegro hand in object grasping tasks, while its temperature regulation functionality was assessed in a material detection task.

## I. INTRODUCTION

Tactile sensors play a crucial role in humanoid robot systems by providing physical contact information that enables adaptive interaction and manipulation. However, their widespread adoption is hindered by two primary challenges. Firstly, the high cost of current commercial sensors[1][2], often exceeding thousands dollar for a hand due to complex designs, prevents mass production and brings high requirements in robotic design and sensor equipment. Second, research has predominantly focused on enhancing single-modal resolution while overlooking the integration of multi-modal sensing[3]. However, these designs ignore the temperature perception, which is essential for achieving human-like interaction, and single-modal force sensing is also easy to be interfered and hard to detect when it occurs.

To address these limitations, we present Bio-Skin, a low-cost, multi-modal thermostatic tactile sensor. It integrates Hall-effect sensors for normal force and piezoresistors for shear force, also can be easily integrated with robotic hand systems such as the Allegro Hand. Innovatively, it is the first tactile sensor to incorporate a thermistor with a heating wire for active temperature sensing and regulation, mimicking the thermostatic capabilities of human skin and allowing robots to accurately perceive object materials in the thermal domain and adapt their interactions accordingly[4]. This multi-modal design also enables a cross-referencing framework to validate force data, significantly improving robustness in complex electromagnetic environments. With a manufacturing cost at less than one-tenth of commercial alternatives, Bio-Skin paves the way for the mass production and adoption of low-cost, robust tactile systems in robotics. This hardware and control code of this work is already open sourced on GitHub.

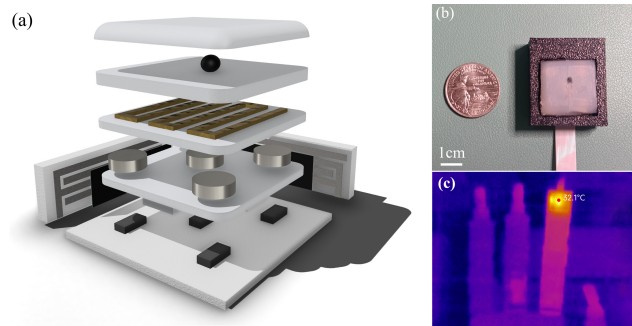

Fig.1. (a). Exploded view of Bio-Skin sensor, showcasing normal force, shear force, and temperature regulation layers with Hall-effect sensors, piezoresistors, magnets, heating and sensing components. (b) Compared with the U.S. quarter dollar, Bio-Skin's silicone body has almost same size with the coin. (c) Bio-Skin maintaining human skin temperature captured by thermal camera.

## II. BIO-SKIN SENSOR DESIGN AND FABRICATION

### A. Sensor Modality

**Normal force:** Hall-effect sensors are widely used for force detections as they can capture subtle magnetic field variations, enabling high-precision force measurement. In this work, the Bio-Skin sensor utilizes five 1-axis Hall-effect sensors to detect the normal planar forces, with five magnets associated with the sensors to provide the magnetic field.

**Shear force:** film piezo resistors are chosen due to their cost-effectiveness and suitability for large-scale production. When shear force is applied, the material deforms, causing a change in resistance. This change is related to the applied force, allowing for sensing shear force.

**Temperature sensing and regulation:** a thermistor is integrated into the top layer to quickly monitor sensor surface temperature, and a heating wire is placed under it to achieve the thermoregulation function and avoid overheating.

### B. Fabrication and Control

The material cost of Bio-Skin is around 10USD, and its fabrication processing can be divided into three steps:

**1) 3D Printing:** Four components are printed: the main silicone body mold, the heating wire fixing mold, the magnets fixing mold, and the fixing frame for shear force. The back of magnets fixing mold is designed with five magnet slots which are used to fix the magnets during molding processing.

**2) Molding:** The molding process involves sequentially molding the magnet, heating and the thermistor layer. The magnet layer requires two molding steps, one for each side. During the second molding step, four springs are installed to provide additional support. Finally, the three layers are molded together to complete the creation of the core body.

**3) Soldering and Assembly:** First, solder the leads of the varistor and the PCB components. Then, affix the conductive film to the core and the varistor around the frame, while simultaneously soldering the leads of the thermistor and the heating wire from the core to the PCB.

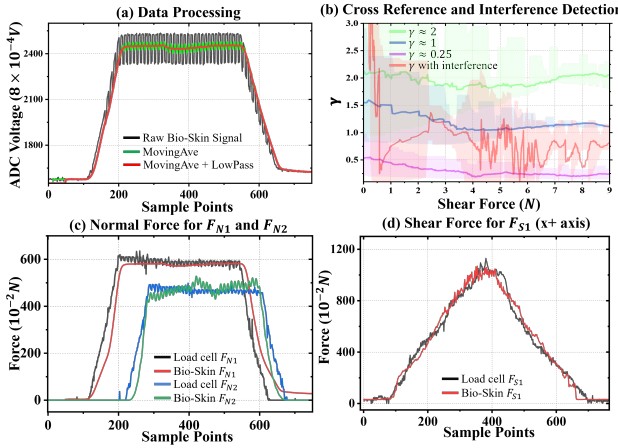

Fig.2. (a) compares the raw data of Bio-Skin and data after filtering. (b) shows the Bio-Skin's interference detection result, red line is detected under changed magnetic field and other 3 is detected under normal environment with different force direction. (c) compares the Bio-Skin's normal force and the ground truth at position 1 and 2, other positions have similar behaviors. (d) compares Bio-Skin's shear force and the ground truth in the positive x-axis, other directions have similar behaviors.

## C. Sensor System Control

A controller samples the Bio-Skin's analog signals and performs analog-to-digital conversion (ADC), measuring five normal forces, four shear force, and one temperature signal for a total of 10 channels. To conserve I/O resources, a separate PCB with a MUX chip is used, with the Bio-Skin connecting to the MUX board via an FPC flex cable, which then connects to the controller. Temperature control is managed by a feedback mechanism where the thermistor's reading adjusts the heating power based on two threshold temperatures: a stop temperature $T_H$ and a full power heating temperature $T_L$, when the temperature between these two temperatures, the power of heating wire will be varied.

## III. EXPERIMENTS AND RESULTS

### A. Cross Reference Validation

Bio-Skin's multi-modal design allows different sensing modalities to cross-reference each other to validate signal fidelity. Thus, in this section, we develop a cross-reference framework between the Hall-effect sensor and piezoresistors in force measurement. According to Newton's law, the normal and shear forces on Bio-Skin are fractions of the direct force. Thus, we can assume a certain relationship exists between these two forces. We define a cross-reference coefficient $\gamma$ by taking the fraction of the calibrated normal and shear force to model a linear relationship:

$$\gamma = F_N/F_S \tag{1}$$

To validate the Cross Reference model, we use the robotic arm applies a certain force on the Bio-Skin surface at three different force directions, so the $\gamma$ ranges from 0.25 to 2, resulting in a total of 30k valid samples. Then, we apply a changing magnetic field around the Bio-Skin to simulate the interference. The results of $\gamma$ are shown in Fig.2b.

### B. Grasping Test

To verify the effectiveness of the super-resolution algorithm and practical application of Bio-Skin, we installed it on an Allegro Hand, which was controlled to grasp a plastic ball and a metal cup while observing the visualization interface

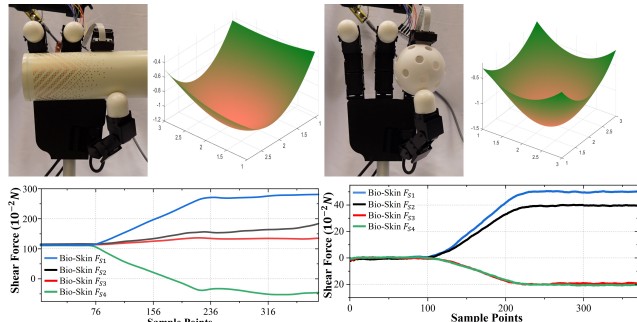

Fig.3. The force visualization of Allegro Hand fingertip with Bio-Skin while grasping a plastic ball (right) and a metal cup (left), showing distinct normal force distribution and shear force changing.

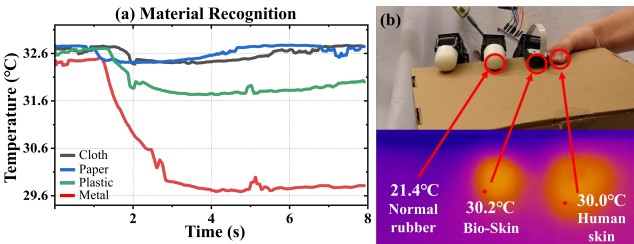

Fig.4. (a) shows the temperature changing after Bio-Skin touch different materials with temperature regulation on. (b) shows temperature retention by comparing Bio-Skin, human skin and normal rubber.

(shown in Fig.3). The results prove that the 3D planner force detection and shear force detection can be used to identify different shapes of objects.

### C. Material Recognition and Temperature Retention

Leveraging its temperature regulation function, Bio-Skin can detect the temperature dissipation rate upon contact, offering an extra modality for material identification due to varying thermal conductivities. To evaluate this, Bio-Skin stabilized at 32°C and then touched different materials at 26°C environment temperature. The result is shown in Fig.4a. Compared to other tactile sensor with temperature sensing function, the heating function of Bio-Skin produces a larger temperature gap between the sensor and environment, resulting in more sensitive measuring of temperature change.

To demonstrate another potential benefit of Bio-Skin's temperature regulation function, we controlled the Allegro Hand to grasp a piece of cardboard for 10 seconds while a human subject simultaneously did the same task. Thermal imaging was then used to observe the retained temperature on the cardboard, as shown in Fig.4b, proving Bio-Skin's ability to mimic the thermal print of human skin.

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
