# OpenReview forum: "Bio-Skin: A Cost-Effective Thermostatic Tactile Sensor with Multi-Modal Force and Temperature Detection"
_IEEE.org/IROS/2025/Workshop/Tactile_Sensing — IROS 2025 Workshop Tactile Sensing Poster_

### Official Review · Reviewer_mVLs · 2025-09-13
**A review for Bio-Skin**

**Rating:** 6
**Confidence:** 3

**Review:**

This paper proposes a novel multimodal tactile sensor that integrates a heating function and a thermistor, enabling better detection of material types. The logic is clear, and the experiments are thorough, but the following limitations remain:

Some existing studies have successfully used Hall effect sensors and magnetic layers to detect 2D shear force effectively. Why is the bar-shaped piezoresistive-based approach for 2D shear force detection proposed in this paper superior to using Hall effect sensors? What are its specific advantages?

The paper lacks explicit data quantifying the extent of improvement in various performance metrics. It is recommended to more intuitively demonstrate the sensor’s performance advantages.

The thermistor appears to be arbitrarily placed inside the sensor without a dedicated positioning or securing mechanism. Does the placement of the thermistor affect sensor performance?

---

### Official Review · Reviewer_RREu · 2025-09-18
**Review report for Bio-skin**

**Rating:** 6
**Confidence:** 4

**Review:**

This work proposes a low-cost tactile sensor for dual-modal sensing of three-axis forces and temperature. The manuscript is well-written, and the topic is hot indeed. Some issues are listed as follows:
1. Thin-film piezoresistance is a typical route to detect normal force. Further array extensions have also been reported for sensing shear forces. The introduction of Hall units seems to lack sufficient motivation, which also complicates the data processing.
2. In Fig.3, the measured forces along opposite directions exhibit asymmetrical amplitudes. The difference channel may improve the accuracy of the estimated shear forces.
3. Material identification based on heat transfer is also affected by contact states such as area and roughness. How to ensure the sensing robustness.